# Learning Non-Parametric Invariances from Data with Permanent Random Connectomes

## Abstract

One of the fundamental problems in supervised classification and in machine learning in general, is the modelling of non-parametric invariances that exist in data. Most prior art has focused on enforcing priors in the form of invariances to parametric nuisance transformations that are expected to be present in data. Learning non-parametric invariances directly from data remains an important open problem. In this paper, we introduce a new architectural layer for convolutional networks which is capable of learning general invariances from data itself. This layer can learn invariance to non-parametric transformations and interestingly, motivates and incorporates permanent random connectomes, thereby being called Permanent Random Connectome Non-Parametric Transformation Networks (PRC-NPTN). PRC-NPTN networks are initialized with random connections (not just weights) which are a small subset of the connections in a fully connected convolution layer. Importantly, these connections in PRC-NPTNs once initialized remain permanent throughout training and testing. Permanent random connectomes make these architectures loosely more biologically plausible than many other mainstream network architectures which require highly ordered structures. We motivate randomly initialized connections as a simple method to learn invariance from data itself while invoking invariance towards multiple nuisance transformations simultaneously. We find that these randomly initialized permanent connections have positive effects on generalization, outperform much larger ConvNet baselines and the recently proposed Non-Parametric Transformation Network (NPTN) on benchmarks that enforce learning invariances from the data itself.

## 1 Introduction

**Learning Invariances from Data using Deep Architectures.** The study of machine learning over the years has resulted in the identification of a few core problems that many other problems are compositions of. Learning invariances to nuisance transformations in data is one such task. A class of architectures have been recently proposed that explicitly attempt to *learn* the transformation invariances directly from the data, with the only prior being the structure that allows them to do so. One of the earliest attempts to do this using backpropagation was the SymNet [4], which utilized kernel based interpolation to learn general invariances. Although given the interesting nature of the study, the method was limited in scalability. Spatial Transformer Networks [5] were also designed to learn activation normalization from data itself, however the transformation invariance learned was parametric in nature. A more recent effort was through the introduction of the Transformation Network paradigm [7]. Non-Parametric Transformation Networks (NPTN) were introduced as an generalization of the convolution layer to model general symmetries from data [7]. It was also introduced as an alternate direction of network development other than skip connections, as is common in ResNets, DenseNets and their variants. The convolution operation followed by pooling

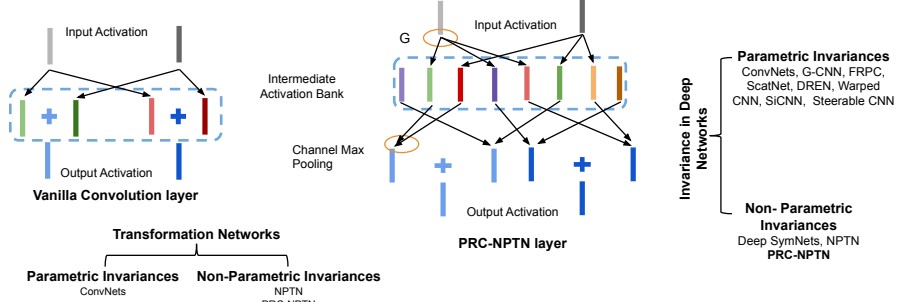

Figure 1: **Left:** Architecture of the vanilla convolution layer. **Left bottom:** Transformation Networks were introduced as a general framework for modelling feed forward convolutional networks. NPTNs and PRC-NPTNs can model non-parametric invariances within the TN framework. **Center:** Architecture of the PRC-NPTN layer. Each input channel is convolved with a number of filters (parameterized by G). Each of the resultant activation maps is connected to a one of the channel max pooling units randomly (initialized once, fixed during training and testing). Each channel pooling unit pools over a fixed random support of a size parameterized by CMP. **Right:** Explicit invariances enforced within deep networks in prior art are mostly parametric in nature. The important problem of learning *non-parametric* invariances from data has not received a lot of attention.

was re-framed as pooling across outputs from the translated versions of a filter. Translation forming a unitary group generates invariance through group symmetry as investigated using computational models of the primary visual cortex [1]. The NPTN framework has the important advantage of learning general invariances without any change in architecture while being scalable. Given this is an important open problem, we introduce an extension of the Transformation Network (TN) paradigm with an enhanced ability to learn non-parametric invariances through permanent random connectivity.

**Relaxed Biological Motivation for Randomly Initialized Connectomes.** Although not central to our motivation, the observation that the cortex lacks precise *local* pathways for back-propagation provided the initial inspiration for this study. It further garnered pull from the observation that random unstructured local connections are indeed common in many parts of the cortex [2, 8]. Though we do not explore these biological connections in more detail, it is still an interesting observation. The common presence of random connections in the cortex at a *local* level leads us to ask: Is it possible that such locally random connectomes improve generalization in deep networks? We provide evidence for answering this question in the positive.

## 2 Permanent Random Connectome NPTNs

**Representation Learning through Pooling.** Over the years, the idea of pooling across transformed features to generate invariance towards that particular transformation has been one of the central tools in algorithm design for invariance properties [3]. Similar ideas have also been explored in a more general setting. For instance, a pose-tolerant feature can be generated by pooling over dot-products of the input face with multiple template faces undergoing pose (and other) variation.

**Invoking Invariance through Pooling.** In previous years a number of theories have emerged on the mechanics of generating invariance through pooling. [1] develop a framework in which the transformations are modelled as a group comprised of unitary operators denoted by $\{g \in \mathcal{G}\}$. These operators transform a given filter $w$ through the operation $gw$[1], following which the dot-product between these transformed filters and an novel input $x$ is measured through $\langle x, gw \rangle$. It is shown by [1] that any moment such as the mean or max (infinite moment) of the distribution of these dot-products in the set $\{\langle x, gw \rangle | g \in \mathcal{G}\}$ is an invariant. These invariants will exhibit robustness to the transformation in $\mathcal{G}$ encoded by the transformed filters in practice, as confirmed by [1].

**The PRC-NPTN layer.** Fig. 1(b) shows the the architecture of a single PRC-NPTN layer. The PRC-NPTN layer consists of a set of $N_{in} \times G$ filters of size $k \times k$ where $N_{in}$ is the number of input channels and $G$ is the number of filters connected to each input channel. More specifically, each of the $N_{in}$ input channels connects to $|G|$ filters. Then, a number of channel max pooling units randomly

---

[1]The action of the group element $g$ on $w$ is denoted by $gw$ to promote clarity.

| Rotation | $0^\circ$ | *** | $30^\circ$ | *** | $60^\circ$ | *** | $90^\circ$ | *** |
|---|---|---|---|---|---|---|---|---|
| ConvNet (36) | $0.70_{\pm 0.03}$ | - | $0.92_{\pm 0.03}$ | - | $1.32_{\pm 0.07}$ | - | $1.93_{\pm 0.02}$ | - |
| ConvNet (36) FC | $0.66_{\pm 0.05}$ | - | $0.80_{\pm 0.03}$ | - | $1.08_{\pm 0.02}$ | - | $1.58_{\pm 0.01}$ | - |
| ConvNet (512) | $0.65_{\pm 0.04}$ | - | $0.80_{\pm 0.02}$ | - | $1.14_{\pm 0.03}$ | - | $1.54_{\pm 0.03}$ | - |
| NPTN (12,3) | $0.68_{\pm 0.06}$ | - | $0.84_{\pm 0.02}$ | - | $1.19_{\pm 0.01}$ | - | $1.64_{\pm 0.02}$ | - |
| PRCN (36,1) | $0.62_{\pm 0.08}$ | $0.62_{\pm 0.06}$ | $0.84_{\pm 0.01}$ | $0.83_{\pm 0.03}$ | $1.17_{\pm 0.05}$ | $1.19_{\pm 0.02}$ | $1.72_{\pm 0.05}$ | $1.73_{\pm 0.06}$ |
| PRCN (18,2) | $0.61_{\pm 0.02}$ | $0.57_{\pm 0.02}$ | $\mathbf{0.68}_{\pm \mathbf{0.02}}$ | $0.73_{\pm 0.02}$ | $\mathbf{0.93}_{\pm \mathbf{0.04}}$ | $0.99_{\pm 0.04}$ | $\mathbf{1.24}_{\pm \mathbf{0.01}}$ | $1.33_{\pm 0.02}$ |
| PRCN (12,3) | $\mathbf{0.58}_{\pm \mathbf{0.03}}$ | $0.62_{\pm 0.04}$ | $0.72_{\pm 0.02}$ | $0.74_{\pm 0.02}$ | $0.95_{\pm 0.01}$ | $1.04_{\pm 0.04}$ | $1.28_{\pm 0.01}$ | $1.33_{\pm 0.01}$ |
| PRCN (9,4) | $0.63_{\pm 0.02}$ | $0.62_{\pm 0.04}$ | $0.75_{\pm 0.02}$ | $0.77_{\pm 0.02}$ | $0.99_{\pm 0.03}$ | $1.05_{\pm 0.03}$ | $1.31_{\pm 0.03}$ | $1.40_{\pm 0.03}$ |
| Translations | 0 pixels | *** | 4 pixels | *** | 8 pixels | *** | 12 pixels | *** |
| ConvNet (36) | $0.69_{\pm 0.04}$ | - | $0.72_{\pm 0.01}$ | - | $1.22_{\pm 0.02}$ | - | $4.43_{\pm 0.05}$ | - |
| ConvNet (36) FC | $0.60_{\pm 0.02}$ | - | $0.64_{\pm 0.01}$ | - | $0.88_{\pm 0.05}$ | - | $3.49_{\pm 0.11}$ | |
| ConvNet (512) | $0.63_{\pm 0.02}$ | - | $0.64_{\pm 0.01}$ | - | $1.00_{\pm 0.02}$ | - | $3.56_{\pm 0.04}$ | - |
| NPTN (12,3) | $0.66_{\pm 0.02}$ | - | $0.64_{\pm 0.02}$ | - | $1.09_{\pm 0.04}$ | - | $4.19_{\pm 0.04}$ | - |
| PRC-NPTN (36,1) | $0.65_{\pm 0.02}$ | $0.65_{\pm 0.05}$ | $0.58_{\pm 0.01}$ | $0.61_{\pm 0.04}$ | $1.02_{\pm 0.03}$ | $1.00_{\pm 0.04}$ | $3.85_{\pm 0.11}$ | $3.83_{\pm 0.10}$ |
| PRC-NPTN (18,2) | $\mathbf{0.59}_{\pm \mathbf{0.07}}$ | $0.59_{\pm 0.03}$ | $\mathbf{0.52}_{\pm \mathbf{0.03}}$ | $0.58_{\pm 0.02}$ | $\mathbf{0.80}_{\pm \mathbf{0.03}}$ | $0.88_{\pm 0.05}$ | $\mathbf{3.23}_{\pm \mathbf{0.03}}$ | $3.34_{\pm 0.06}$ |
| PRC-NPTN (12,3) | $0.63_{\pm 0.02}$ | $0.66_{\pm 0.08}$ | $0.55_{\pm 0.02}$ | $0.59_{\pm 0.01}$ | $0.84_{\pm 0.04}$ | $0.89_{\pm 0.03}$ | $3.35_{\pm 0.04}$ | $3.52_{\pm 0.12}$ |
| PRC-NPTN (9,4) | $0.65_{\pm 0.02}$ | $0.69_{\pm 0.03}$ | $0.56_{\pm 0.03}$ | $0.56_{\pm 0.03}$ | $0.88_{\pm 0.02}$ | $0.97_{\pm 0.02}$ | $3.49_{\pm 0.46}$ | $3.69_{\pm 0.08}$ |

Table 1: **Individual Transformation Results:** Test error statistics with mean and standard deviation on MNIST with progressively extreme transformations with a) **random rotations** and b) **random pixel shifts**. $***$ indicates ablation runs without any randomization *i.e.* without any random connectomes (applicable only to PRC-NPTNs). For PRC-NPTN and NPTN the brackets indicate the number of channels in the layer 1 and $G$. ConvNet FC denotes the addition of a 2-layered pooling $1 \times 1$ pooling network after every layer. Note that for this experiment, CMP=$|G|$. Permanent Random Connectomes help with achieving better generalization despite increased nuisance transformations.

select a fixed number of activation maps to pool over. This is parameterized by Channel Max Pool (CMP). Note that this random support selection for pooling is the reason a PRC-NPTN layer contains a permanent random connectome. These pooling supports once initialized do not change through training or testing. Once max pooling over CMP activation maps completes, the resultant tensor is average pooled across channels with a average pool size such that the desired number of outputs is obtained. After the CMP units, the output is finally fed through a two layered network with the same number of channels with $1 \times 1$ kernels, which we call a pooling network. This small pooling network helps in selecting non-linear combinations of the invariant nodes generated through the CMP operation, thereby enriching feature combinations downstream.

**Invariances in a PRC-NPTN layer.** Recent work introducing NPTNs [7] had highlighted the Transformation Network (TN) framework in which invariance is generated during the forward pass by pooling over dot-products with transformed filter outputs. A vanilla convolution layer with a single input and output channel (therefore a single convolution filter) followed by a $k \times k$ *spatial* pooling layer can be seen as a single TN node enforcing translation invariance with the number of filter outputs being pooled over to be $k \times k$. It has been shown that $k \times k$ spatial pooling over the convolution output of a single filter is an approximation to channel pooling across the outputs of $k \times k$ translated filters [7]. The output $\Upsilon(x)$ of such an operation with an input patch $x$ can be expressed as $\Upsilon(x) = \max_{g \in \mathcal{G}} \langle x, gw \rangle$ where $\mathcal{G}$ is the set of filters whose outputs are being pooled over. Thus, $\mathcal{G}$ defines the set of transformations and thus the invariance that the TN node enforces. In a vanilla convolution layer, this is the translation group (enforced by the convolution operation followed by *spatial* pooling). An NPTN removes any constraints on $\mathcal{G}$ allowing it to approximately model arbitrarily complex transformations. A vanilla convolution layer would have *one* filter whose convolution is pooled over spatially (for translation invariance). In contrast, an NPTN node has $|\mathcal{G}|$ *independent* filters whose convolution outputs are pooled across *channel* wise leading to general invariance.

A PRC-NPTN layer inherits the property from NPTNs to learn arbitrary transformations and thereby arbitrary invariances using $\mathcal{G}$. As Fig. 1(b) shows, individual channel max pooling (CMP) nodes act as NPTN nodes sharing a *common* filter bank as opposed to independent and disjoint filter banks for vanilla NPTNs. This allows for greater activation sharing, where transformations learned from data through one subset of filters can be used for invoking similar invariances in a parallel computation path. This sharing and reuse of activation maps allows for higher parameter and sample efficiency. As we find in our experiments, randomization plays a critical role here, allowing for a simple and quick approximation to obtaining high performing invariances.

## 3 Empirical Evaluation and Discussion

**Efficacy in Learning Arbitrary and Unknown Transformations Invariances from Data.** We evaluate on one of the most important tasks of any perception system, *i.e.* being invariant to nuisance transformations *learned* from the data itself. We benchmark our networks based on tasks where nuisance transformations such as large amounts of in-plane rotation and translation are steadily increased, with no change in architecture whatsoever. For this purpose, we utilize MNIST where it is straightforward to add such transformations without any artifacts. We benchmark on such a task as described in [7] and for fair comparisons, we follow the exact same protocol. We train *and* test on MNIST augmented with progressively increasing transformations *i.e.* **1)** extreme random translations (up to 12 pixels in a 28 by 28 image), **2)** extreme random rotations (up to $90°$ rotations). *Both* train and test data were augmented leading to an increase in overall complexity of the problem. No architecture was altered in anyway between the two transformations *i.e.* they were not designed to specifically handle either. The same architecture for all networks is expected to learn invariances directly from data unlike prior art where such invariances are hand crafted in [6].

For this experiment, we utilize a two layered network with the intermediate layer 1 having up to 36 channels and layer 2 having exactly 16 channels for all networks (similar to the architectures in [7]) except a wider ConvNet baseline with 512 channels. All ConvNet, NPTN and PRC-NPTN models have the similar number of parameters (except the ConvNet with 512 channels). For PRC-NPTN, the number of channels in layer 1 was decreased from 36, through to 9 while $|G|$ was increased in order to maintain similar number of parameters. All PRC-NPTN networks have a two layered $1 \times 1$ pooling network with same number of channels as that layer. For a fair benchmark, Convnet FC has 2 two-layered pooling networks with 36 channels each. Average test errors are reported over 5 runs for all networks.

**Discussion.** We present all test errors for this experiment in Table. 1[2]. It is clear that as more nuisance transformations act on the data, PRC-NPTN networks outperform other baselines with the same number of parameters. In fact, even with significantly more parameters, ConvNet-512 performs worse than PRCN-NPTN on this task for all settings. Since the testing data has nuisance transformations similar to the training data, the only way for a model to perform well is to learn invariance to these transformations. It is also interesting to observe that permanent *random* connectomes do indeed help with generalization. Indeed, without randomization the performance of PRCN-NPTNs drop substantially. The performance improvement of PRC-NPTN also increases with nuisance transformations, showcasing the benefits arising from modelling such invariances.

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

---

[2]We display only the (12, 3) configuration for NPTN as it performed the best.

