# OpenReview forum: "Learning Non-Parametric Invariances from Data with Permanent Random Connectomes "
_NeurIPS.cc/2019/Workshop/Neuro_AI — Submitted to Real Neurons & Hidden Units @ NeurIPS 2019_

### Official Review · AnonReviewer1 · 2019-09-26
**Poorly written and of unclear importance**

**Clarity:** 1

**Category:**

AI->Neuro

**Clarity Comment:**

The paper is written in a dense and difficult to follow style. Part of this is due to the heavy reliance on previous NPTN literature. But it is also due to use of jargon and poorly defined parameters. Examples include G (if it is a number what is |G| needed for?) and CMP. The authors should strive to provide an intuitive description of their results. The diagram in figure 1 does not do a good job of describing the architecture. No general discussion of the results is provided.

**Evaluation:**

2: Poor

**Importance:**

2: Marginally important

**Importance Comment:**

The paper focuses on the topic of learning non-parametric invariances using randomly wired networks. A network architecture is proposed that extends previous approaches and improves performance on an MNIST dataset with various transformations applied to it. The results are rather preliminary and their importance is difficult to assess due to the poor presentation of the paper.


**Intersection:**

2: Low

**Intersection Comment:**

The connection to neuroscience is quite loose, as the authors acknowledge. The authors speculate that local random connectivity is present in the brain, but beyond that little discussion of the biological relevance of the results is made.

**Rigor Comment:**

The results appear to provide an improvement over the previous NPTN work. However, because only one benchmark is used, it is difficult to assess the generality of the results.


**Technical Rigor:**

2: Marginally convincing

---

### Official Review · AnonReviewer2 · 2019-09-26
**Rough writing somewhat obscures what is likely an important and interesting method for forming invariant representations**

**Clarity:** 2

**Comment:**

The exposition needs to be cleaned up. Figure 1 in particular needs to be expanded to include more models and more details. The authors should consider keeping only one of the organization trees in Figure 1 since the two feel redundant, or find a way to combine them.

The buildup from the theory, to Transformation Networks, then to Non-Parametric Transformation Networks, then to Permanent Random Connectome Non-Parametric Transformation Networks, and finally to comparisons with convolution neural networks, should probably have happened in a more streamlined, linear way.

Regarding the biological motivation for the fixed connections, this point could be strengthened somewhat by describing how the max pooling could be implemented by biology. The theory (as developed in [1]) seems to hold for averaging as well as max pooling, which may be more biologically feasible. In general I think the authors don't give themselves enough credit with the connection to biology (where the computationally beneficial aspects of random connections is already being discussed, i.e. for dimensionality expansion), and they could have laid out the connections more clearly.

And finally, making the point that random fixed connections are important/useful, beyond showing simulation results, would strengthen the work considerably.

The paper does a less-than-stellar job of making the case that a presentation of the work at the workshop would leave attendees with a basic understanding. That said, the paper has a great deal of potential, and could contribute significantly to both fields if clarity issues are resolved.

**Category:**

Common question to both AI & Neuro

**Clarity Comment:**

Overall the paper suffers from messy organization, difficult-to-digest expositions about the differences of at least four closely related models, some missing details, and some lack of motivation and intuition. Some of this is understandable due to the intrinsically complex nature of the work, but it seems that with more time and polish the paper could be improved a great deal (and still fit in four pages).

Below are some specific examples of clarity issues.

It would be helpful if the "random unstructured local connections" as seen in cortex were defined more precisely. Do these connections not change as the animal learns tasks while other connections do change? Do these connections map together different "filters", as in the authors' proposed model?

There is a bundle of small issues with the writing. For instance, the acronym PRC is defined in the abstract but not in the main text. In Table 1, the labels in the caption are missing in the table itself. The label for the PRC-NPTN networks is different in the rotation table vs the pixel translation table.

The organization of the paper gets in the way of its clarity. For instance, Transformation Networks are introduced in Section 1, but it isn't until Section 2 that the underlying theory is referenced (reference [1]). As far as I can tell Transformation Networks are a direct application of this theory to deep neural networks. This connection isn't made as explicit as it could have been.

The architecture could have been made clearer if Figure 1 had shown an example of a (Non-Parametric) Transformation Network layer, as well as a standard convolution layer with max pooling, to compare with the Permanent Random Connectome Non-Parametric Transformation Network.

While some intuition is provided for why random connections are advantageous over the standard Non-Parametric Transformation Network layers, a more thorough discussion of this important point would have been very helpful. Why is it helpful to max pool across different filters?

**Evaluation:**

4: Very good

**Importance:**

4: Very important

**Importance Comment:**

The learning of invariances is a key problem in both machine and biological intelligence, and any progress made to understand it is of high importance. While the less-than-perfect clarity of the work makes it a little harder to ascertain the authors' success at making progress on this problem, it seems to me as though it is a solid step in the right direction. I might rate this as a 4.5 if I could.

**Intersection:**

4: High

**Intersection Comment:**

While the authors don't do very much to explain the connection to biology and confess that this isn't a strong motivator for them, I believe that the connection is actually fairly strong. Success in their models suggests roles for random connections in the brain. Their results suggest potential improvements to state-of-the-art performance in artificial neural networks, since convolutional layers in very deep architectures could conceivably be swapped out by the layers proposed here. As such, the results are interesting both to the neuroscientist as well as the AI researcher.

I feel that putting more effort into making the connection to biology could easily increase this score by a point.

**Rigor Comment:**

When it comes to building invariances, an important issue is being able to learn with fewer training examples than an architecture that doesn't have as much invariance-building capability. Here the authors present "errors" of their trained models without elaborating much further. It would have been helpful if the authors had discussed the training more (such as if they train until the error no longer decreases) and maybe shown the accuracy through training.

There are other details that aren't explained. For instance, an important aspect of the model are fixed random connections, but it isn't discussed if these connections are weighted or not. These missing details don't seem to be essential to me.

**Technical Rigor:**

4: Very convincing

---

### Decision · Program_Chairs · 2019-10-01

**Decision:**

Reject

**Comment:**

Unfortunately, we had more submissions than we could accept and based on the review process, we have decided not to accept your submission.  Nevertheless, thank you for your submission and interest in our workshop.